# Rapid Tryptophan Assay as a Screening Procedure for Quality Protein Maize

**DOI:** 10.3390/molecules29184341

**Published:** 2024-09-12

**Authors:** Gabi Drochioiu, Elena Mihalcea, Jeanclaude Lagobo, Catalina-Ionica Ciobanu

**Affiliations:** 1Faculty of Chemistry, Alexandru Ioan Cuza University of Iasi, 11 Carol I, 700506 Iasi, Romania; lenuta_mihalcea@yahoo.com (E.M.); lagobojeanclaude@gmail.com (J.L.); 2Environmental Science Laboratory (LSE), Nangui-Abrogoua University, Abidjan 02 BP 801, Côte d’Ivoire; 3Integrated Centre of Environmental Science Studies in the North East Region—CERNESIM Centre, Institute of Interdisciplinary Research, Alexandru Ioan Cuza University of Iasi, 11 Carol I, 700506 Iasi, Romania; catalina.ciobanu@uaic.ro

**Keywords:** tryptophan determination, *Zea mays* L., quality protein maize, protein determination, simultaneous analysis

## Abstract

Tryptophan is an essential amino acid deficient in cereals, especially maize. However, maize (*Zea mays* L.) is the main source of protein in some developing countries in Africa and Latin America. In general, the nutritional profile of cereals is poor, because they are deficient in essential amino acids such as tryptophan and lysine due to a relatively higher proportion of alcohol-soluble proteins. Quality protein maize (QPM) has been developed through genetic manipulation for the nutritional enrichment of maize to address these problems. Nevertheless, methods for protein, lysine and tryptophan are time-consuming and require relatively large amounts of samples. Therefore, we have advanced here a simple, cheap, fast, reliable and robust procedure for the determination of protein and tryptophan in the same biuret supernatant, which can also be used for chemical characterization of other cereals. Samples of 50 mg maize ground to pass through a 0.1 mm screen were sonicated for 5 min. in eppendorf vials with 1.5 mL of a biuret reagent each. After centrifugation and protein determination by biuret, 0.2 mL of supernatant was treated with 0.8 mL of a tryptophan reagent. Both total protein and tryptophan can be determined in microplates at 560 nm to speed up the measurements. The main advantage of the new micro-method is the rapid estimation of the nutrient quality of maize samples by a single weighing of a small amount of valuable plant materials.

## 1. Introduction

The poor protein quality of normal maize seeds is caused by zein proteins, which lack tryptophan and lysine residues [1,2,3]. Consequently, zein determination has been previously studied to determine the nutritional quality of opaque-2 and common maize [4,5,6,7]. However, nutrient-rich maize genotypes by incorporating crtRB1 and o2 genes associated with increased levels of β-carotene, lysine, and tryptophan have been studied and developed [8]. Then, several naturally occurring mutants have been identified that confer high levels of lysine and tryptophan to maize, namely opaque-2, opaque-6, opaque-7, floury-2, and floury-3 [9,10]. For instance, opaque-2 (o2) mutants undergo a decrease in γ-zein and an increase in lysine-rich non-zein proteins through a phenomenon known as proteome rebalancing [9,11,12]. Thus, quality protein maize (QPM) was developed in the late 1960s [13,14] to produce 70% to 100% more lysine and tryptophan than common maize varieties [15,16,17]. In addition, QPM lines have both high protein and tryptophan levels and can be used as parents for highly nutritious hybrids [18]. Nutritional scoring of QPM in nutrition trials reveals its nutritional superiority over non-QPM varieties for human and animal consumption [13]. Some authors have estimated that the nutritional quality of QPM proteins approaches that of cow’s milk proteins [14]. The potential of QPM to help reduce protein deficiency in Africa has also been highlighted [19,20].

In order to determine the nutritional quality of cereals, including maize, it is necessary to determine both the crude or pure protein content and the content of essential amino acids required by the human or monogastric animal body [3,21,22]. Tryptophan is an essential amino acid required by the body, which can only be obtained through daily food intake. The other essential amino acid, lysine, which is also found in a small proportion in maize and other cereals, is actually in a 4:1 ratio with tryptophan [23,24,25]. For this reason, we consider that the determination of tryptophan by its reaction with glyoxylic acid [26] would be preferable for the estimation of cereal nutritional quality in crop breeding and the selection of QPM maize lines. In addition, the methods of analysis proposed until recently are based on the separate determination of proteins from the determination of tryptophan or lysine [18,27,28].

Protein hydrolysis performed in the presence of Ba(OH)_2_ was found to yield the highest and least dispersed values for the tryptophan content of plant materials [29]. An electrochemical sensor based on a glassy carbon electrode for tryptophan determination via differential pulse voltammetry was proposed [30]. A glassy carbon electrode modified with nanocomposites of green and chemically synthesized copper oxide-polyaniline was thus applied to determine tryptophan in pineapple [31]. Accurate determination of tryptophan in food was also done by fluorescence analytical technique [32,33]. Many other methods have been employed to detect tryptophan, such as ultra-high performance liquid chromatography with electrospray ionization, cryoelectron microscopy, and spectroscopic techniques like fluorescence microscopy-mass spectroscopy with a triple quadrupole [34,35]. However, the methods for quantification of protein-bound amino acids of plant materials often involve a complicated hydrolysis set-up and several hydrolysate processing steps [36]. Therefore, particularly in plant-breeding work, such as the production of QPM hybrids, rapid analytical methods are required. They should also be high-yielding, low in plant material and sensitive to relevant levels of proteins and essential amino acids of interest.

In this work, we proposed the simultaneous determination of the protein in maize samples and the tryptophan contained in it. This eliminates time-consuming double weighing of plant material samples. To further increase the speed of determination of these two components, we used a microplate spectrophotometer. Indeed, a microplate reader with 96 wells, together with instruments capable of recording measurements from them, allows a large number of samples to be determined rapidly [37]. Here, the method for the determination of tryptophan using a residual biuret solution from a protein assay was tested in comparison with the colorimetric method described in the literature [27]. We also discussed our results in tryptophan determination for their potential utility in maize-breeding programs. In addition, the essential differences between conventional spectrophotometric measurements and those using microplate readers were discussed.

## 2. Results

### 2.1. Tryptophan Reaction

To study the reaction of tryptophan with a reagent containing sulfuric acid, acetic acid, and acetic anhydride, a standard solution of 0.1 mM tryptophan (20.4 μg/mL) was prepared [38]. Thus, 20.4 mg of tryptophan was dissolved in 10 mL of an alkaline alcoholic solution. Then, 1 mL of a stock solution was diluted with the same alkaline alcoholic solution to 100 mL. From the resulting 0.1 mM tryptophan alkaline solution, a volume of 1.5 mL was sonicated with 50 mg of insoluble copper phosphate for 15 min. Eppendorf vials were subsequently centrifuged at 15,000 rpm for 5 min, and 1 mL of the supernatant was treated with 4 mL of a tryptophan reagent in tubes with vigorous shaking. The absorbance spectra were plotted against the control made with reagents only in the range of 300–700 nm in 1 cm quartz cuvettes (Figure 1). The molar absorptivity of tryptophan (molecular weight: 204.23) at 558 nm was 4770 L mol^−1^ cm^−1^. In addition, the reagent components (the blank) did not interfere with the determination of tryptophan at the maximum wavelength (558–560 nm) or even in the range from 450 to 700 nm.

We also tested the determination of tryptophan in a microplate reader at 600 nm (Table 1). When pipetting six different 0.25 mL samples of 100 μM tryptophan in a biuret solution, the mean absorbance measured in a microplate reader was 0.359 ± 0.0098 AU (blank: 0.060–0.065; samples: 0.409–0.438). The limit of detection, LOD = 3.3·s_x_/S and the limit of quantification, LOQ = 10·s_x_/S, respectively, were LOD = 0.016 and LOQ = 0.049 μM, respectively. Here S was the slope of the regression line (S = 280.48). The following regression equation was calculated, T = 280.48·TA − 0.4339 (R^2^ = 0.998), where T = tryptophan concentration as μM, and TA, tryptophan absorbance). These results demonstrated the high sensitivity of the proposed micro-method.

We determined the selectivity, linearity, accuracy, precision, robustness, detection limit, and quantitation limit of the proposed method for tryptophan determination. We also calculated RMSE values (0.0109) as well as CV = 0.0489 (RSD = 1.36). A calibration curve was plotted against a reagent blank in the concentration range of 0–100 μM (Figure 2). The regression model is a linear one in which the relationship between tryptophan concentration (C) and the absorbance at 560 nm (A_560_) is a straight line, A_560_ = 0.0036C, where 0.0036 is the slope of the line. Linearity of the calibration curve was expressed through the coefficient of correlation, r = 0.995, and the coefficient of determination, R^2^ = 0.99, although r is not an appropriate measure for the linearity. Therefore, using residual plots is a simple way to check linearity [39]. We thus observed that the A560 values were normally distributed for a linear model, and no curves were observed to suggest a lack of fit due to a nonlinear effect.

In this work, the assessment of nutritive quality of opaque-2 (QPM) and normal maize samples was performed by simultaneous determination of crude protein and tryptophan as described below, compared with separate determination of crude protein by the micro-Kjeldahl method and tryptophan by a method proposed by Nurit et al. [27]. For this purpose, we selected 25 maize samples, 12 normal maize samples, and 13 QPM-type samples, with tryptophan content determined after 16 h of papain hydrolysis in the range of 0.37–0.82% in crude protein (Appendix A). A strong relationship was observed between tryptophan values determined from biuret supernatants and those from enzyme hydrolysates (R^2^ = 0.983; r = 0.991). The linear regression equation was y = 1.1086x − 0.0561, where y is %tryptophan in crude protein determined in the biuret supernatants and x is the values of the enzyme-based method. However, the two types of maize samples displayed tryptophan values in two distinct groups with inflection points. For this reason, the Bland–Altman analysis was conducted to compare the obtained results and present quantified measures to decide whether the new method is approvable or not [40]. Appendix A shows the agreement between two tryptophan measurements, according to the limits of agreement quantified for the two datasets. A one-sample *t*-test was performed to calculate the mean bias and its SD. For our dataset, the mean difference was found as 0.0108 with an SD of 0.0298. Therefore, in our data set, the upper limit can be calculated using mean + 1.71 × SD (0.0108 + 1.71 × 0.0298 = 0.0618), and the lower limit can be calculated using mean − 1.71 × SD (0.0108 + 1.71 × 0.0298 = −0.04027). The appropriate statement used in the manuscript can be as follows: the Bland–Altman plot showed that the mean ± SD between the first and the second tryptophan levels was 0.0108 ± 0.0298% tryptophan, and the limits of agreement were −0.04027 and 0.0618 (Appendix A).

Therefore, the two methods can be employed interchangeably, as the limits vary within 0.1% tryptophan.

In addition, intraday and interday precisions of this analytical method were found to be reliable based on RSD% (<2%). The values of 1.36% (n = 6) and 1.64% (n = 6) were identified for intraday and interday precisions, respectively, confirming that the method is sufficiently precise.

### 2.2. Protein Determination

The new variant of the biuret micro-method, which was described in Section 4.4.1, total protein determination, was applied for the determination of proteins. The crude protein was thus determined on the basis of a calibration curve using the values of biuret absorbance versus crude protein content of maize samples analyzed by the micro-Kjeldahl method (Figure 3). Indeed, the total protein determined by the biuret method was similar to the crude protein content measured by the micro-Kjeldahl method (Figure 3). Ultrasonic stirring enhanced the biuret reaction in the presence of alkaline alcoholic solutions and insoluble copper phosphate. Ethanol prevented the opalescence generated by the starch of the maize samples, while phosphate powder was the source of copper ions that formed a biuret complex with the peptide bonds of proteins under alkaline conditions. The regression equation was B = 1.0664∙K − 0.7313 (R^2^ = 0.9755; r = 0.987), where B is the percent of crude protein determined by the biuret method, and K, the micro-Kjeldahl values.

The new variant of the biuret method, using insoluble copper phosphate instead of copper sulfate and sodium and potassium tartrate, has been advanced for the determination of proteins in maize samples and for obtaining biuret supernatants for tryptophan quantification. The spectra of the biuret absorbance and the standard curves at various wavelengths of 1–5 mg mL^−1^ bovine serum albumin (BSA) are shown in Appendix A.

The results showed that it is feasible to determine proteins both at 545 nm, where biuret absorbance is maximal, and at 560 nm in a microplate reader. However, if BSA is used as a standard to plot the calibration curve to determine protein in real samples, then the procedure should be the same. In our study, 1 mL of 1–5 mg mL^−1^ BSA was treated with 0.5 mL of an alkaline alcoholic solution, whereas proteins from real samples were extracted in only 1 mL of an alkaline alcoholic solution. In daily determinations, we preferred to analyze 50–100 samples by the biuret method described here, to which we add 4–5 samples with crude protein content determined by the micro-Kjeldahl method and covering a wide range of values (e.g., 8–15% crude protein). These 4–5 samples were sufficient to plot calibration curves each day. The crude protein content was reported on dry matter, determined in an oven at 105 °C for 24 h.

### 2.3. Characterization of Real Samples

Crude protein contents of 25 maize samples were determined by the micro-Kjeldahl (Appendix A) and were found in the concentration range from 7.72% to 15.97%. Increased precision and repeatability were also found in the determination of proteins in maize using the proposed biuret method. A highly significant correlation was observed between the biuret absorbance (BA) of the samples at 545 nm in 1 cm quartz cuvettes and the crude protein content determined by the micro-Kjeldahl method, r = 0.987 with the regression equation, BA = 0.0567·CP + 0.0307, where CP is the percentage of crude protein in the dry matter.

Appendix A also shows that normal maize might be high in protein, but its crude protein contained high amounts of zein, lysine and tryptophan-deficient protein. For example, sample HT-17/81 had 15.97% crude protein and 6.68% zein. Nevertheless, the hybrid HT Sv 17/81 was retained in the breeding program for use in polygastric animal feed. Zein values were particularly low for QPM-type maize, ranging from 1.10% to 2.42%, while normal maize had significantly higher values of this parameter (2.22–6.68%).

A strong relationship was noticed only between zein and crude protein in normal maize and not in QPM. In addition, the QPM samples contained low amounts of zein. Moreover, a negative correlation was found between lysine and zein in maize samples studied (Appendix A). Although, the correlation coefficient was r = −0.823, it is however difficult to differentiate the QPM samples from the normal ones, when compared with tryptophan determination (Appendix A).

Although QPM varieties showed on average relatively lower crude protein values, some opaque-2 maize varieties nevertheless achieved particularly high values for this parameter (O-26/86: 13.94%), suggesting the possibility of increasing the protein content of opaque maize while maintaining high values for lysine and tryptophan. Thus, by applying both the classical methods of lysine and tryptophan determination and that proposed in this paper, QPM samples can be easily differentiated from normal maize (Appendix A). Indeed, the lysine content was higher in opaque maize samples than in normal maize (lysine in total protein: 3.10–3.77% and 1.88–2.43%, respectively). The lysine content of normal maize still showed a decrease with the increasing protein level of the analyzed maize plants (Appendix A). However, tryptophan levels followed the same pattern, so that the more rapid and simpler determination of tryptophan can replace more complicated and time-consuming lysine determination. Basically, the sensitive increase in the protein content of the plants did not represent a sensitive change in the lysine content, so that the increase in the protein index was realized only at the expense of the increase in the zein content of the samples, which was independent of lysine content.

In this work, we used a residual biuret supernatant after biuret reaction and treated one volume of the supernatant with four volumes of tryptophan reagents. In fact, the biuret solution can be recovered from protein quantification by the biuret method and used for the determination of tryptophan and lysine. In addition, tryptophan absorbance divided by the biuret absorbance values could be a parameter of choice for assessing the nutritional quality of maize samples.

The strong correlation between the values of the percentage of tryptophan in protein determined by papain hydrolysis and those obtained by the proposed micro-method (r = 0.982), as well as the slope of the regression line (1.108) indicated that essentially all tryptophan was available for the reaction, as shown in Appendix A. The statistical analysis revealed that there was no significant difference between the proposed method and the methods proposed by Nurit et al. [27] and by Hernández and Bates [38] for the determination of tryptophan in maize samples (*p* > 0.05).

Tryptophan content in normal maize seeds commonly ranges from 0.2% to 0.5% of the total protein content, while it typically varies from 0.5 to 1.1 in opaque-2 maize [27]. In our study, tryptophan values ranged from 0.37% to 0.82% in crude protein and 35.09–100.36 mg% in dry matter (Appendix A). Indeed, normal maize samples showed less than 0.5% tryptophan in DM.

A strong correlation was found between tryptophan and lysine values (r = 0.979***), demonstrating once again the possibility of assessing the quality of maize by determining its tryptophan content (Appendix A). As expected, a negative correlation was determined between the zein content of the analyzed samples and the tryptophan content of their dry matter (r = −0.491), suggesting that zein content is less dependent on the nutritive quality of maize samples. The following regression equation was generated: y = −6.9527x + 83.755 (R^2^ = 0.2419), where y is tryptophan level expressed as mg% in DM and x is the level of zein expressed as % in DM.

## 3. Discussion

The novelty and originality of these studies consists in the use of a rapid assessment of the nutritional quality of maize protein by the determination of tryptophan and crude protein in the same biuret supernatant. In addition, this simultaneous determination of maize components could be of great interest both for breeding work and for the normal characterization of maize or other cereals.

Lysine, tryptophan, and threonine are limiting amino acids in humans and non-ruminants. Tryptophan and lysine could be determined quickly, simply and accurately from the residual biuret solution from protein determination by the biuret method. However, only tryptophan and crude protein were determined here, given the strong correlation between the two essential amino acids, which allows the nutritional quality of maize to be assessed by knowing the content of only one of them. Although we have advanced this method based on simultaneous determination of protein and tryptophan in the biuret supernatant of maize samples, the proposed procedure can be applied not only for the identification of QPM hybrids, but also for the measurement of protein and tryptophan in normal maize. Indeed, opaque-2 mutants of maize induce increased levels of lysine and tryptophan in maize endosperm of QPM [9]. The development of QPM hybrids through advanced breeding approach like molecular marker-assisted breeding was adopted. It could solve the issue related to protein deficiency in developing countries.

Existing methods for estimating the nutrient quality of QPM and normal maize samples are time-consuming and plant-material-intensive. On the contrary, our innovation eliminates these drawbacks and allows for the rapid and reliable determination of both tryptophan and proteins in biuret supernatants obtained from a single weighing. In addition, the Bradford method is significantly affected by the presence of salts, while the biuret assay remains stable [41]. The biuret reaction is mostly applied to determine the amount of soluble protein in a solution [42]. However, we have used insoluble copper phosphate which released small amounts of soluble copper ions capable of extracting proteins from maize meal. These form the biuret complex with the proteins in alkaline medium, the absorbance of which is proportional to the protein content of the maize samples.

Protein quality was previously expressed as the content of tryptophan in endosperm proteins [43]. According to classical methods, samples were defatted; then protein and tryptophan were determined separately. However, we show in this paper that it is possible to identify QPM varieties by analyzing the whole seeds and not only the endosperms. In addition, corn oil in the maize samples does not interfere with the measurements according to our proposed method. Moreover, only one weighing of a small amount of plant material to be analyzed is required, as this micro-method has been specifically dedicated to breeding work as well as to the use of microplate readers.

Only common, inexpensive reagents and glassware are needed. Besides, no special training is necessary, except the usual reading of colored solutions using a spectrometer or microplate reader. Moreover, the biuret method is quicker and simpler than that of the Kjeldahl method and can be recommended as a replacement for it in breeding work.

If needed, the pass-length of any aqueous solution in a microplate well can be determined by comparing the absorbance of that solution at a known pass-length (e.g., 1 cm) with the absorbance in the microplate well. Nevertheless, correction for the passage length in the microplate wells is not necessary, because the calibration curves are drawn under the same conditions as the samples. In addition, results are obtained more quickly and can be easily transferred to an Excel spreadsheet to be automatically calculated as protein or tryptophan concentrations.

Previously, we also used the simplified methods with good results to differentiate high protein and tryptophan samples from normal maize.

Future research should test the micro-method on a large number of samples of maize and other breeding cereals and to improv it.

## 4. Materials and Methods

### 4.1. Reagents

The reagents and solvents used here were of analytic purity, and the aqueous solutions were prepared with pure water (18.2 MΩ·cm) obtained from a Millipore water purification system (Millipore, Bedford, MA, USA).

Corn zein protein and bovine serum albumin (BSA) were purchased from Sigma-Aldrich (Saint Louis, MO, USA) and were used for calibration curves and other analytical studies. A 4 mg mL^−1^ stock solution of zein in 70% ethanol and a 5 mg mL^−1^ stock solution of BSA were prepared for drawing the calibration curves by appropriate dilution. Solvents for degreasing flour samples and protein extraction, such as ethanol, acetone, and petroleum ether, were purchased from Merck (Darmstadt, Germany) and were utilized without further purification. Petroleum ether was the option of choice to defat maize flours by Soxhlet Traditional Solvent Extraction (TSE).

An alkaline-alcoholic solution (AA or biuret reagent) was prepared by dissolving 20 g of KOH in about 200 mL of milliQ water, followed by adding 500 mL of 95% (*v*/*v*) ethanol and making up to 1 L with water.

Papain solution. Papain technical powder was dissolved in a 0.1 acetate buffer solution with pH 7 and a final concentration of 4 mg mL^−1^ papain.

Tryptophan reagent. Equal volumes of acetic solution and 30 N sulfuric acid were mixed one hour before use. The acetic solution consisted of 27 mg FeCl_3_·6H_2_O dissolved in 0.5 mL H_2_O over which 100 mL of glacial acetic acid containing 4 mL acetic anhydride was added. The 30 N sulfuric acid was prepared by adding dropwise 81.5 mL of concentrated sulphuric acid (98%) over 19 mL of water, under stirring and cooling with tap water, followed by making up to 100 mL with water.

### 4.2. Plant Material to Be Analyzed

The research samples consisted of normal and opaque-2 maize kernels with a moisture content of about 12% from several maize hybrids and inbred lines taken from the Plant Genetic Resources Bank of Suceava, Romania. When classical analytical methods were applied for comparative studies, the kernels were ground to a fine powder and the resulting flour was defatted using petroleum ether as solvent in a Soxhlet extractor. Dry matter (DM) was determined in a stove at 105 °C for 24 h.

### 4.3. Instrumentation

Maize kernels were milled to a fine powder using a laboratory electric cereal mill (SAMAP Mod F100, Andolsheim, France) with adjustable millstones. The resulted flours were screened with an analytical vibratory sieve shaker (Retsch Gmbh, Haan, Germany) to obtain flour fully passing a certain sieve (for example, 100 μm mesh sieve).

An ultrasonic bath (J.P. Selecta Ultrasons system, 40 kHz; Barcelona, Spain) was used for both ultrasonic extraction and biuret reaction. Centrifugation at 15,000–18,000 rpm of the solvent mixtures with the working samples was carried out using a Hettich Mikro 22R centrifuge (Tuttlingen, Germany). The UV-vis absorption spectra were taken on a Biochrom Libra S35 PC UV–visible spectrophotometer (Cambridge, England) in quartz cuvettes of 10 mm (1 mL volume) in the range from 200 to 900 nm. In addition, a Modulus™ microplate reader (Turner Biosystems, Sunnyvale, CA, USA) was used. All measurements were carried out at room temperature.

### 4.4. Procedures

#### 4.4.1. Total Protein Determination

Each 50 mg duplicate of maize flour was treated under shaking with 1 mL of alkaline-alcoholic solution (AA) in a plastic eppendorf vial. Approximately 50 mg of copper phosphate powder was then added and the mixture was sonicated for 30 min. If necessary, other ratios between the amount of flour and the volume of the alkaline-alcoholic solution can be used. The eppendorf vials were centrifuged at 15,000 rpm, and the absorbance spectra were read in the range from 200 to 900 nm; however, the 545 nm absorbance (or the 560 nm absorbance in the microplate reader) of the supernatant was of interest for our experiments, as it was proportional to the protein content of the samples. Biuret absorbance values were compared with those obtained by the micro-Kjeldahl method. In addition, a calibration curve with micro-Kjeldahl values was made for some representative maize samples (high, medium, and low protein contents or biuret absorbances).

#### 4.4.2. Tryptophan Determination in the Biuret Supernatant

After protein determination by the biuret method, 200 μL of biuret supernatant was taken and placed in an eppendorf vial, after which 800 μL of tryptophan reagent was pipetted. The mixture is shaken vigorously and incubated at 70 °C for 15–30 min to perform the color reaction. Vials were cooled to room temperature, and the absorbance of the purple-violet solution is read at 558 nm in 1 cm glass cuvettes against a control using reagents only.

#### 4.4.3. Using the Microchip Reader

After incubation and centrifugation, 200 μL of biuret supernatant was transferred into a 96-well plate, and the absorption signal proportional to the protein content was detected at 560 nm with a Modulus™ microplate reader. The results were expressed as mean relative light units (RLUs) and compared with those obtained by the normal UV–vis spectrophotometry.

Similarly, 250 μL of each violet-violet solution following the reaction of tryptophan with glyoxylic acid was pipetted into a 96-well plate, and the absorbance value proportional to the tryptophan content was also detected at 560 nm. The results were expressed as mean relative light units (RLUs) and compared with those obtained by the normal UV–vis spectrophotometry. Calibration curves were plotted with 0–5 mg mL^−1^ BSA for protein and 0–50 μg mL^−1^ tryptophan.

#### 4.4.4. Other Analytical Methods

The crude protein content of defatted maize meal was determined by the micro-Kjeldahl method [44], and the values obtained are given in Appendix A.

The tryptophan content was also quantified by the papain hydrolysis method indicated by Hernández and Bates in 1969 [38], as well as by Nurit in 2009 [27]. However, it was not expressed as percentage of tryptophan in endosperm protein, but in crude protein of the whole maize kernel (Appendix A). The sample of 100 mg defatted finely powdered maize flour was mixed with 4 mL of papain solution, incubated for 16 h at 63 ± 2 °C, cooled and centrifuged at 5000 rpm for 5 min. Then, 1 mL of hydrolyzate was transferred to a solution of ferric chloride and sulfuric acid, incubated at the 65 °C for 15 min, allowed to cool, and read at 560 nm to calculate the amount of tryptophan relative to the protein on a standard curve 0–50 μg mL^−1^.

Lysine was evaluated by the method of Tsai et al. (1975), which was modified by Villegas (1984) [45,46]. From the samples of maize hydrolyzed with papain at 65 ± 2 °C for 16 h, after cooling to room temperature, 1 mL was placed in a copper carbonate solution, vortexed for 5 min and centrifuged at 5000 rpm for 5 min. Afterwards, 1 mL of the supernatant was mixed with 0.1 mL of a 2-chloro-3,5-dinitropyridine solution. The mixture was then stirred vigorously and allowed to stand for 2 h and shaken every 30 min; hydrochloric acid was added, and the mixture was again stirred until homogenized. Subsequently, an ethyl acetate extraction solution was added, stirred by inversion 10 times, and the upper phase was extracted, repeating three times and read at 390 nm against a blank. The lysine content was calculated from a standard curve and plotted against protein.

In our experiments, zein in the defatted maize samples was extracted with a 70% aqueous ethanol solution [47,48,49]. Duplicates of 100 mg of maize meal were placed in test tubes over which 2 mL of 70% ethanol was pipetted and ultrasonic stirred for 30 min. 1 mL of zein extract was treated with 0.5 mL of a biuret reagent and 50 mg of insoluble copper phosphate, and after sonication for 5 min and centrifugation at 15,000 rpm, the supernatant was read at 545 nm in a spectrophotometer. The calibration curve was plotted with alpha zein as a standard.

#### 4.4.5. Simplified Methods

For measurements in the field or for poorly equipped laboratories, we provide simpler or naked eye methods. Thus, 100 mg of maize flour is mixed with 3 mL of a biuret reagent in 180/18 mm test tubes fitted with stoppers, and the mixture is shaken. Add about 50 mg of insoluble copper phosphate to the mixture, and shake the mixture from time to time. After approximately one hour, the contents of the tubes are filtered through Whatman No 1 filter paper, and the absorbance is measured at 545 nm using a spectrophotometer or with a colorimeter at 540 nm in 1 cm glass or plastic cuvettes. Plot a calibration curve with representative values obtained with the micro-Kjeldahl method or bovine serum albumin. From the biuret filtrate, take 1 mL which is pipetted into another test tube over which 4 mL of tryptophan reagent is pipetted. Shake the tubes and maintain them at 65 °C for 15 min. The absorbance of the violet- purple solutions is measured at 550–560 nm against a control made with reagents alone. The calibration curve is plotted with tryptophan in the concentration range of 0–50 μg mL^−1^.

Alternatively, a naked eye method may be used, which involves using a series of 10 test tubes for the calibration curves and visually comparing the sample to be analysed with them.

### 4.5. Statistics

The coefficient of variation (CV), the coefficient of determination (R^2^), the relative standard deviation (RSD%), the root mean square error (RMSE) [50], the limit of detection (LOD), and the limit of quantification (LOQ), as well as the correlation coefficient (r), were calculated to compare the results of the proposed method with those of conventional methods. The Bland–Altman analysis [40] was also applied to investigate the agreement between the proposed micromethod and the reference one based on enzymatic hydrolysis [27] in the determination of tryptophan levels in different maize samples.

## 5. Conclusions

The method for estimating the nutritional quality of maize samples based on the rapid determination of protein and tryptophan content in a biuret solution is characterized by simplicity, speed, and increased productivity. The method has the advantage to eliminate time-consuming double weighing of plant material samples.

Crude protein from maize samples was ultrasonic-assisted extracted in an alkaline-alcoholic solution in the presence of insoluble copper phosphate, centrifuged and determined in a supernatant at 560 nm as a biuret complex in a microplate reader. The values obtained by the proposed micro-biuret test were in full agreement with those of the micro-Kjeldahl method. In addition, the modified biuret micro-method proved to be simple, productive and reliable.

The tryptophan content determined by the proposed micro-method was also found to be in good agreement with that obtained by reference methods in the literature. Data were achieved with high degree of precision and sensitivity. Relative standard deviation was 1.36% (n = 6).

Analyzing real samples of normal and QPM maize, we observed a strong correlation only between the crude protein contents of normal maize and zein. Therefore, it is difficult to differentiate the two types of maize solely by their zein content values. On the other hand, tryptophan values that are strongly correlated with those of lysine allowed for good differentiation of QPM samples from normal ones, which is an important advantage of the method proposed here.

Application of the proposed method could considerably increase productivity in maize breeding and reduce the requirement of valuable plant materials.

## 6. Patents

Part of this study is based on our own earlier patent [51]. Thus, we improved the nutritional quality of some normal corn by eliminating a large part of zein, as evidenced by the analysis of the resulting flours by chemical reference methods.

## Figures and Tables

**Figure 1 molecules-29-04341-f001:**
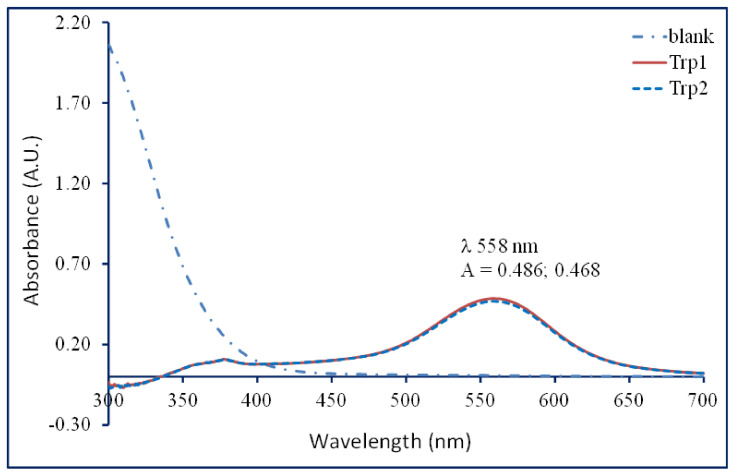
Typical UV-vis spectra of tryptophan (1 mL; 0.1 mM) derivative formed in the reaction with glyoxylic acid in a tryptophan reagent (4 mL).

**Figure 2 molecules-29-04341-f002:**
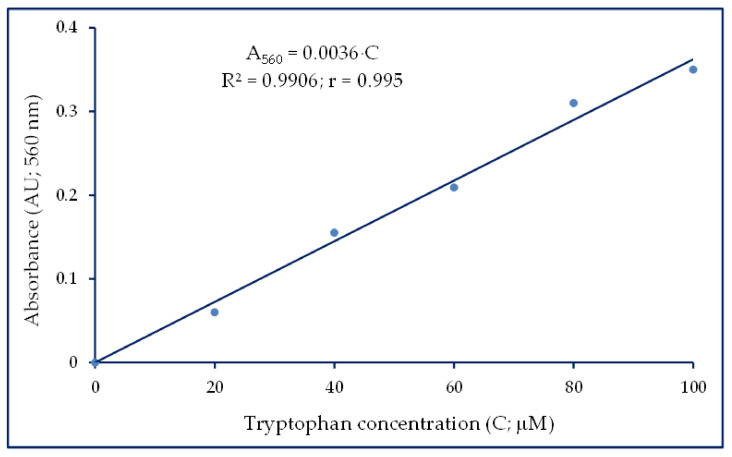
Standard curve of tryptophan using a microplate reader (0–100 µM; volumes of 250 µL in duplicate).

**Figure 3 molecules-29-04341-f003:**
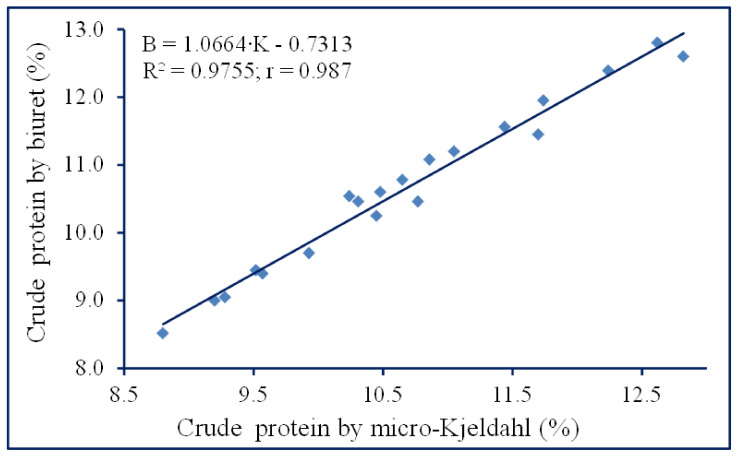
Relationships between the crude protein contents of 23 different maize samples determined by the micro-Kjeldahl method and the protein values determined from the corresponding absorbance values of the biuret supernatant.

**Table 1 molecules-29-04341-t001:** Tryptophan absorbance measured with a microplate reader at 560 nm.

Volume ^1^	Blank1	Blank2	Trp1	Trp2	Trp1−Blank1	Trp1−Blank2	Mean
100 μL	0.072	0.072	0.224	0.217	0.151	0.145	0.148
100 μL	0.073	0.074	0.221	0.215	0.149	0.141	0.145
250 μL	0.064	0.065	0.421	0.438	0.358	0.373	0.365
250 μL	0.062	0.060	0.411	0.409	0.349	0.349	0.349

^1^ C_Tryptophan:_ 100 mM (Trp1 & Trp2).

## Data Availability

All data in this paper are available from the corresponding author’s (GD) Database.

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
