# Peer review of "Rapid Tryptophan Assay as a Screening Procedure for Quality Protein Maize"

_molecules, 2024, doi:10.3390/molecules29184341_

Round 1

Reviewer 1 Report

Comments and Suggestions for Authors

Your scientific contribution is interesting, but not suitable for publication from a methodological point of view.

Agreement and correlation are widely used terms that evaluate the relationship between variables. Although they are similar and related, they represent completely different notions of association. Assessing agreement between variables assumes that the variables measure the same construct, whereas correlation of variables can be assessed for variables that measure completely different constructs.

Below are some comments:

Line 97, page 3: Replace »11.03 – 20.99«, with »11.03–20.99«. Correct this error throughout the paper.

Figure 1: You have an influential point in the data set, and the Pearson correlation coefficient is sensitive to such points.

Line 145, page 4: It is better to say strong than high correlation between…

Line 160, page 5: The word »close correlation« is not appropriate in statistics.

Figure 3: For such data, Pearson's correlation coefficient is not a suitable method as the R is overestimated."

You need to rewrite chapter 4.5 and use more appropriate methods such as Bland-Altman analysis, an improved Bland-Altman method for concordance assessment, RMSE…

Reviewer 2 Report

Comments and Suggestions for Authors

This paper presents a rapid method for determining tryptophan content, which offers certain advantages over traditional methods. However, the writing logic of this paper is somewhat disorganized, with some sections lacking logical coherence. I consider the paper should focus more on discussing the advantages of the method rather than merely presenting the measurement results.

In the abstract, the authors provide detailed procedural information, which I find unnecessary. Instead of detailing the experimental procedures, the abstract should emphasize the advantages of this method compared to traditional methods and its accuracy in measuring tryptophan.

In lines 88-92, Table 1 presents the nutritional content of various maize varieties, but it does not sufficiently support the conclusion that "this is strongly correlated with high amounts of zein."

In sections 2.1 and 2.2, the paper should highlight the measurement results compared to other methods rather than focusing on the protein content of different maize varieties, which is unrelated to the main theme of the paper.

In lines 169-170, the authors praise the advantages of the method using several adjectives, but the subsequent discussion lacks sufficient evidence to support these claims.

In lines 231-232, the authors mention that the method requires further improvement, but they did not previously discuss any limitations of the method, making this statement seem abrupt.

In lines 364-365, the phrase "saving time and valuable plant material for breeding" needs clarification. How does omitting a single sample weighing save time and resources needed for breeding?
